# Genetic Diversity in Phytoplasmas from X-Disease Group Based in Analysis of *idpA* and *imp* Genes

**DOI:** 10.3390/microorganisms13051170

**Published:** 2025-05-21

**Authors:** Florencia Ivette Alessio, Vanina Aylen Bongiorno, Carmine Marcone, Luis Rogelio Conci, Franco Daniel Fernandez

**Affiliations:** 1Instituto de Patología Vegetal (IPAVE), Centro de Investigaciones Agropecuarias (CIAP), Instituto Nacional de Tecnología Agropecuaria (INTA), Córdoba X5020ICA, Argentina; floralessio17@gmail.com (F.I.A.); bongiorno.vanina@inta.gob.ar (V.A.B.); conci.luis@inta.gob.ar (L.R.C.); 2Unidad de Fitopatología y Modelización Agrícola (UFYMA), Consejo Nacional de Investigaciones Científicas y Técnicas (CONICET), Córdoba X5020ICA, Argentina; 3Department of Pharmacy, University of Salerno, 84084 Fisciano, Italy; cmarcone@unisa.it

**Keywords:** membrane proteins, X-disease, phylogeny, selection, *imp*, *idpA*

## Abstract

Phytoplasmas of the X-disease group (16SrIII) are economically significant pathogens in South America, causing severe crop losses. Traditional classification based on the 16S rRNA gene has limitations in resolving closely related strains, prompting the exploration of alternative markers. This study focuses on the immunodominant membrane proteins imp and idpA, which exhibit high variability and play crucial roles in host–pathogen interactions. Through molecular characterization of *imp* and *idpA* genes in 16SrIII subgroups, we identified significant genetic diversity and distinct evolutionary pressures. The *imp* gene, under positive selection, showed high variability in its hydrophilic extracellular domain, suggesting adaptation to host immune responses. In contrast, *idpA* exhibited strong negative selection, indicating functional conservation. Phylogenetic analyses revealed that *imp* and *idpA* provide higher resolution than the 16S rRNA gene, enabling finer differentiation within subgroups. These findings highlight the potential of *imp* and *idpA* as complementary markers for phytoplasma classification and diagnostics.

## 1. Introduction

Phytoplasmas are cell wall-less bacteria with an obligate parasitic lifestyle, associated with diseases in hundreds of plant species [1,2]. These pathogens are transmitted between plants by sap-sucking insect vectors, including leafhoppers, planthoppers, and psyllids [3]. The classification of phytoplasmas has traditionally relied on sequence analysis of the 16S rRNA gene, initially using restriction fragment length polymorphism (RFLP) of a 1.25 kb fragment to define ribosomal (16Sr) groups and subgroups. While this system, established in the 1990s, remains useful for grouping (with >37 groups and 150 subgroups described to date), species delineation originally employed a 97.5% 16S rRNA sequence identity threshold [2,4,5,6]. Despite its widespread adoption, the 16Sr classification system for phytoplasmas exhibits several notable limitations. These include its reliance on restriction fragment length polymorphism (RFLP) analysis of the 16S rRNA gene, which captures variation only at selected restriction sites and overlooks the broader sequence context [7]. Moreover, the presence of intragenomic heterogeneity—multiple divergent copies of the 16S rRNA gene within a single genome—can complicate taxonomic assignments and lead to inconsistent subgroup identification [8]. Additionally, the system provides insufficient resolution for inferring phylogenetic relationships among closely related strains, which limits its utility in detailed evolutionary or epidemiological studies [9]. These drawbacks have prompted researchers to adopt more robust methods for phytoplasma classification. For instance, multilocus sequence typing (MLST), which uses multiple single-copy housekeeping genes, offers greater discriminatory power. Studies have effectively differentiated strains using markers such as *secY*, *rp*, and *leuS* in combination with the 16S rRNA gene [10,11,12]. The development of genome-enabled multilocus sequence analysis (MLSA) has further enhanced classification accuracy by allowing researchers to design markers based on full genome comparisons and to assess their performance in genome-scale phylogenies [13]. This approach has proven especially powerful in well-studied groups like 16SrI, which benefit from a wealth of genomic data [14]. As sequencing technologies advance, a shift toward whole-genome-based taxonomy is increasingly feasible. Genomic metrics such as average nucleotide identity (ANI) are now recognized as more reliable tools for defining species boundaries, offering a stronger alternative to 16S rRNA gene-based methods [15,16,17].

Phytoplasmas of the 16SrIII group are agriculturally significant in South America, infecting a wide range of plants, including weeds, vegetables, fruit trees, and staple crops [18,19,20]. The 16SrIII-J subgroup is particularly relevant in South America, associated with X-disease in Argentina, Southern Brazil, and Chile, causing significant crop losses [18,21,22,23,24,25]. Another important subgroup is 16SrIII-B, found in a variety of hosts such as China tree, peach, plum, tomato, and cassava, in Argentina, Brazil, and Paraguay [26,27,28]. Other subgroups within 16SrIII include 16SrIII-W, associated with *Heterothalamus alienus* causing leaf reduction, and 16SrIII-X, found in *Erigeron bonariensis* and lettuce, showing witch’s broom and malformed flowers [18,29]. Additionally, 16SrIII-L has been linked to frogskin disease in cassava, where symptoms primarily affect tubers [30]. The diversity of these subgroups, including 16SrIII-B, 16SrIII-J, and 16SrIII-X, highlights their adaptability to various hosts and environments, posing ongoing challenges for disease management. Their widespread distribution and economic impact in South America emphasize the need for continued research and improved diagnostic and management strategies. Immunodominant membrane proteins (IDPs) of phytoplasmas, such as Imp, IdpA, and Amp, are highly abundant proteins located on the outer membrane, interacting directly with plant and insect hosts. These proteins play a key role in host–pathogen interactions, including adhesion to insect vectors and plant colonization [29]. IDPs exhibit high sequence variability due to positive selection, reflecting their adaptation to evading host immune responses and optimizing transmission [31,32,33]. This variability, combined with their membrane localization and abundance, makes them ideal candidates for molecular markers of diversity. Recombinant expression of IDPs can generate specific antibodies, enabling their detection and characterization across phytoplasma strains [33,34]. Given the agricultural importance of the X-disease group in South America and the limited genomic data available for 16SrIII subgroups, this study focuses on the molecular characterization of the single-copy membrane protein genes *imp* and *idpA* in phytoplasmas of subgroups 16SrIII-J, 16SrIII-B, and 16SrIII-X from Argentina to improve phylogenetic resolution and enhance understanding of their host–pathogen interactions.

## 2. Materials and Methods

### 2.1. Plant Samples

Total DNA from eleven isolates previously reported in Argentina and belonging to the X-disease group (subgroups 16SrIII-J, 16SrIII-B, and 16SrIII-X) were used for molecular assays (Table 1). DNA was isolated from leaf midribs according to the Doyle and Doyle [35] CTAB protocol. The detection and classification of phytoplasmas were performed by direct and nested PCR amplification of 16S rRNA using the primer pairs P1/P7 [36] and R16F2n/R16R2 [37], respectively. The RFLP profiles of nested amplicons were analyzed as described in previous works [18,24]. Although the initial screening included samples previously associated with the 16SrIII-W subgroup, repeated attempts to amplify DNA from these samples were unsuccessful despite the use of different extraction methods and PCR protocols. As a result, 16SrIII-W was excluded from this study, and analyses were focused on the 16SrIII-B, 16SrIII-J, and 16SrIII-X subgroups.

### 2.2. Identification of imp and idpA Homologues

The coding sequences for the *imp* and *idpA* genes were identified in published phytoplasma genomes belonging to the X-disease group (16SrIII) available in the NCBI database (NCBI: txid85623) (Table 2). To identify these genes, BLASTp (version 2.16.0) searches were conducted using the *imp* (AP314487.1) and *idpA* (AP31480.1) sequences from the WX phytoplasma as queries. For phytoplasmas lacking annotations in the NCBI database, functional annotations were performed using Prokka (v1.14.5) [38]. Subsequently, BLASTp searches were carried out using Geneious R.10 software (Biomatters Ltd., Auckland, New Zealand) to identify homologous sequences.

### 2.3. PCR Amplification of imp and idpA Genes

Specific PCR primers were designed to clone the complete coding sequences of the imp and idpA proteins. The genomic context of each gene (*imp* and *idpA*) was manually examined, and conserved regions across all X-disease genomes were identified for primer design. The final primer sets were validated using Primer3 2.3.7 within Geneious R.10 software. The PCR amplification of genomic fragments was performed in a 40 µL reaction volume containing 100 ng of DNA, 0.4 mM of each primer, 200 µM of each dNTP, 1 U of GoTaq^®^ DNA polymerase, 1× polymerase buffer (Promega, Madison, WI, USA), and sterile water. The thermal cycling conditions for *imp* and *idpA* genes were as follows: an initial denaturation at 94 °C for 3 min, followed by 35 cycles at 94 °C for 1 min, 54 or 58 °C for 1 min (for *imp* or *idpA*, respectively), and 72 °C for 1 min 30 s, with a final extension at 72 °C for 8 min. PCR amplicons were visualized by electrophoresis on 1% agarose gels stained with GelRed^®^ and imaged under UV light.

### 2.4. Cloning and Sequencing of imp and idpA Genes

The PCR amplicons were purified using commercial columns (PBL, Quilmes, Argentina) and cloned into pGEM^®^-T Easy Vector System I (Promega, USA) following the manufacturer’s recommendations. *E. coli* DH5α competent cells were used for transformation. For each sample, three clones were bidirectionally sequenced (3× minimum coverage per base) using an automated Sanger DNA sequencer service (Macrogen, Seoul, Republic of Korea). Consensus sequences were assembled using Geneious R.10 software and deposited in the NCBI GenBank. Open reading frames were estimated using ORF Finder and annotated using BLASTp (nr, BLOSUM62, word size 6). The deduced amino acid sequences of Imp and IdpA proteins were analyzed using SignalP 5.0 [39] for prediction of signal peptide sequences. Transmembrane helix domains were predicted using TMHMM v2.0 [40].

### 2.5. Genetic Diversity

The nucleotide and amino acid identity of the sequences were calculated using MAFFT v7.505 [41]. The number of polymorphic sites and nucleotide diversity (Pi, Jukes and Cantor) were evaluated using DnaSP v6.0 [42]. For the target genes (*imp* and *idpA*), synonymous (*dS*) and non-synonymous (*dN*) nucleotide substitution values were calculated using the Nei-Gojobori (Jukes-Cantor) method implemented in MEGA v7.0 [43]. Codon-based Z-Test of Neutrality was used to reject the null hypothesis of strict neutrality (*dN* = *dS*). For positive selection, *dN*/*dS* values > 1 and *p*-value < 0.05 were considered significant [44]. A maximum likelihood analysis of natural selection codon-by-codon was conducted using the HyPhy software package (2.3.10) [45] implemented in MEGA v7.0. For the 16S rRNA gene, nucleotide diversity (π) and segregating sites were calculated from gap-free alignments using MEGA 7.0. As a non-coding sequence, selection pressure analysis (*dN*/*dS*) was not performed. All positions containing gaps or missing data were eliminated prior to analysis.

### 2.6. Phylogenetic Analysis of X-Disease Phytoplasmas

Phylogenetic relationships were evaluated using individual and concatenated alignments of the *imp*, *idpA*, and 16S rRNA genes from nine X-disease phytoplasma genomes (NCBI) and six isolates sequenced here (Table 2). The individual and concatenated alignments were generated with MAFFT (L-ins-I; k = 2). Maximum likelihood trees were reconstructed in IQ-TREE [46] with automatic model selection and 1000 ultrafast bootstrap replicates. 

## 3. Results

### 3.1. Identification of imp and idpA ORFs in X-Disease Genomes

In this analysis, the sequences of nine genomes described in Table 2 were used. Of the nine phytoplasma genomes analyzed, two belong to subgroup 16SrIII-J, two to subgroup 16SrIII-B, two to subgroup 16SrIII-F, and three to subgroup 16SrIII-A (Ca. Phytoplasma pruni). The complete sequences of the *imp* and *idpA* genes were identified in all isolates. In addition, the presence of the *amp* gene was investigated using the *amp* protein sequence of the ‘*Candidatus Phytoplasma asteris*’ strain AYWB (GenBank accession number: WP_011412878.1) as a query. No *amp* homologs were identified in any of the genomes analyzed.

### 3.2. PCR Amplifications and Sequencing

For the *imp* gene, a primer pair (imp-Fw: 5′-ATCTCGTCCTCTTAAACCGCATCC-3′; imp-Rv: 5′-AGACTCTTAACTGGCAACG-3′) was designed to amplify a ~1.0 kb fragment covering the complete ORF of the imp protein. Similarly, for the *idpA* gene, a primer pair (idpA-Fw: 5′-CCCTTCTGCTCCGCCAATTA -3′; idpA-Rv: 5′- TTGCCGAGCAAAAGAGCAAT -3′) was designed to amplify a ~1.4 kb fragment encompassing the entire ORF of the idpA protein. The PCR amplification of 1.0 kb (*imp* gene) and 1.4 kb (*idpA* gene) was successfully achieved in eight out of eleven strains, respectively, as shown in Table 1. No amplification was observed in the negative controls (control mix and healthy samples). The complete coding sequences for the imp and idpA proteins were obtained from the FbtWY, GDIII, BellVir, CicWB (subgroup 16SrIII-J), CaesLL (subgroup 16SrIII-B), BidPhy, and LWB (subgroup 16SrIII-X) strains. For the ChTDIII strain (subgroup 16SrIII-B) isolate, although amplicons were obtained for both genes, the sequences obtained from the genome were used for the analysis. The size of the coding region of the *imp* and *idpA* genes analyzed in this work falls within the expected range for these genes according to previous reports [47]. For the *imp* gene, the open reading frame lengths ranged from 522 to 546 bp among the strains studied. In contrast, the *idpA* gene showed greater size variation, ranging from 864 bp (e.g., CX isolate) to 987 bp (e.g., JR1, WX, and MW1 isolates).

### 3.3. Sequences Homology and Predicted Protein Structure

The analysis of amino acid sequences from all strains sequenced in this study, as well as those retrieved from NCBI, reveals a conserved domain structure consistent with previous findings for the imp [31,48] and idpA protein [49,50]. In the case of imp, a typical hydrophilic C-terminal domain was observed in all sequences, while the N-terminal region encoded a hydrophobic domain (putative transmembrane helix) of approximately 40 amino acids, which likely serve as an anchor to the phytoplasma cell membrane (Appendix A). We did not infer the presence of a signal peptide or putative cleavage motif in any of the analyzed sequences. Regarding similarity, we found 100% identity values among all isolates of subgroup 16SrIII-J, except for the Vc33 isolate (Chile, periwinkle), which showed a value of 97.73% compared to the other members of the subgroup. On the other hand, isolates from subgroup 16SrIII-A exhibited amino acid identity values ranging from 97.73% to 91.57%, while those from 16SrIII-F had an identity of 90.29%. In 16SrIII-B, the phytoplasmas ChTDIII (Argentina, China tree) and CaesLL (Argentina, *C. gillensi*) exhibited identity values of 71.84% with each other. However, when compared to the MA1 (Italy, daisy) isolate, the identity value dropped to 51.98% for ChTDIII and 55.93% for CaesLL. For the sequences of 16SrIII-X, the identity values ranged from 54.4% compared to the VAC isolate to 44.75% compared to the CaesLL isolate (Appendix A). The overall homology was lower in the exposed hydrophilic region, with a pairwise identity value of 61.50%, compared to the overall sequence with a value of 66.40%.

For all analyzed strains, the structure of idpA protein showed a large central hydrophilic region flanked by two hydrophobic regions (putative transmembrane helices) near the C- and N-terminus (Appendix A). A signal peptide (35 aa) was inferred in all sequences, while no putative cleavage motif was identified. Regarding similarity, we found that the identity values among all isolates of 16SrIII-J varied between 100% and 87.04% (Vc33 vs. all). In subgroup 16SrIII-A, IdpA identity values ranged from 100% (JR-1 vs. PR2021) to 63.83% (PR2021 vs. CX), while within 16SrIII-B, these values ranged from 86.77% (ChTDIII vs. GDIII) to 67.48% (CaesLL vs. MA-1). For 16SrIII-F, the identity was 81.71% (MW-1 vs. MA-1), and for 16SrIII-X, it was 100% (BidPhy vs. LWB) (Appendix A). Like the findings for imp, the global homology was lower in the exposed hydrophilic region, with a pairwise identity value of 68.69%, compared to the overall sequence with a value of 73.20%.

### 3.4. Genetic Diversity

In this study, 16 sequences of the imp protein coding genes were used for the selection pressure analysis. Multiple alignments of 471 positions (157 codons) were evaluated, and *dN*/*dS* was calculated for each codon. Eighty codons showed *dN*/*dS* > 0 values, indicating that the protein would be under positive selection pressure (*dN*/*dS* general= 3.474 *p* = 0.01), of which seventy were found to encode for amino acids in the hydrophilic region exposed (Appendix A). The same analysis was performed with 16 sequences of the gene encoding for the idpA protein, where 253 codons were evaluated. Of these, 115 showed *dN*/*dS* values greater than 0 (*dN*/*dS* general = −3.090 *p* = 0.002), indicating that the protein could be under selection pressure. However, 50 codons exhibited negative *dN*/*dS* values, and overall, the protein appears to be under negative selection pressure. This suggests that, despite the presence of some codons with positive values, the idpA protein is mainly subjected to negative selection. Within these 253 codons, 104 encode for amino acids in the hydrophilic region exposed. The results of these analyses determined that the highest number of codons with *dN*/*dS* values > 0 occurred in the extracellular region [Table 3], indicating that in both proteins, this domain (hydrophilic) is the most variable. For the 16S rRNA gene, an analysis of 1163 nucleotide positions revealed lower nucleotide diversity (π = 0.014) and fewer segregating sites (94) compared to the protein-coding genes, despite its longer sequence length, reflecting higher conservation in this non-coding ribosomal marker.

### 3.5. Phylogeny Based on 16S rRNA, idpA, and imp Genes

The phylogenetic analysis based on 16S rRNA, *idpA*, and *imp* genes provides complementary insights into the evolutionary relationships among 16SrIII phytoplasma subgroups. The 16S rRNA tree establishes a well-defined taxonomic framework, with 16SrIII-J isolates (BellVir, CicWB, Vc33, FbWY, GDIII) forming a strongly supported monophyletic group (bootstrap = 77%), clearly distinct from 16SrIII-B (ChTDIII, CaesLL, MA-1). Similarly, 16SrIII-A/S (CX, JR-1, PR2021/WX) clusters with strong support (97%), while 16SrIII-F (MW-1, VAC-1) and 16SrIII-X (BidPhy, LWB) are more phylogenetically distinct (Figure 1). However, while the 16S rRNA gene is highly conserved and effective for broad classification, it lacks the resolution to detect functional differentiation or evolutionary adaptation [9]. Notably, the 16SrIII-B and 16SrIII-F subgroups show a degree of association, but their placement remains separate.

The *idpA* gene phylogeny introduces additional differentiation within subgroups. While 16SrIII-J members remain clustered (bootstrap = 95–100%), Vc33 is more divergent, suggesting increased variation in *idpA* compared to the highly conserved 16S sequences (Figure 2A). The 16SrIII-B subgroup is less cohesive in *idpA*, with MA-1 appearing more distantly related to ChTDIII and CaesLL. Additionally, 16SrIII-F (MW-1, VAC-1) now clusters more closely with 16SrIII-B (MA-1), reinforcing a pattern not evident in the 16S-based tree. The 16SrIII-A/S subgroups retain their structures (bootstrap = 100%) but show an increased association with 16SrIII-F. The 16SrIII-X isolates (BidPhy and LWB) remain the most divergent. Overall, *idpA* phylogeny provides higher resolution than 16S rRNA, revealing subgroup differentiation and evolutionary interactions. The *imp* gene phylogeny exhibits the highest degree of sequence divergence, indicating distinct evolutionary pressures acting on this locus (Figure 2B). 16SrIII-J members remain partially clustered but show a looser relationship compared to the previous trees, with Vc33 displaying significant divergence. The 16SrIII-B subgroup (ChTDIII, CaesLL, MA-1) is even more fragmented, with CaesLL and ChTDIII forming a strongly supported clade (bootstrap = 100%), yet at a considerable evolutionary distance from other subgroups. Interestingly, MW-1 and VAC-1 (16SrIII-F) remain associated but are now positioned closer to 16SrIII-B, reinforcing a trend seen in *idpA*. The 16SrIII-A/S subgroups remain well-defined (bootstrap = 100%) but appear more closely linked to 16SrIII-F, suggesting possible functional adaptation. Finally, BidPhy and LWB (16SrIII-X) exhibit extreme divergence, consistent with their distinct evolutionary trajectory. Overall, the *imp* phylogeny highlights strong functional divergence across subgroups, suggesting it is more influenced by selective pressures than either *idpA* or 16S rRNA. Together, these three markers provide a comprehensive view of phytoplasma evolution, with 16S rRNA clarifying taxonomy, *idpA* refining subgroup differentiation, and *imp* revealing a putative adaptive divergence.

The concatenated phylogenetic analysis based on the 16S rRNA, *idpA*, and *imp* genes yielded a final alignment of 2794 base pairs. The resulting tree (Appendix A) consistently supports the monophyly of 16SrIII-J isolates (bootstrap = 100%) while also revealing internal diversification, notably with the divergence of Vc33. The 16SrIII-B subgroup (isolates ChTDIII and CaesLL) is well defined (bootstrap = 100%); however, the MA-1 isolate appears more closely associated with the 16SrIII-F subgroup, a pattern observed consistently across individual and concatenated analyses. Meanwhile, 16SrIII-X isolates (BidPhy, LWB) remain highly divergent across the concatenated data-set (bootstrap = 100%). The concatenated phylogeny thus offers enhanced resolution compared to individual gene trees, delineating subgroup-specific evolutionary patterns while maintaining a conservative interpretation of the observed relationships.

## 4. Discussion

Phytoplasmas belonging to the X-disease group (16SrIII) exhibit remarkable genetic diversity and are widely distributed across South America, affecting multiple plant species [18,19,20,24]. In Argentina, several subgroups within this group have been reported, including 16SrIII-B, J, X, and W [18,24]. Traditionally, diversity within this group has been assessed using the 16S rRNA gene; however, this approach has limitations such as incomplete sequence coverage, low phylogenetic resolution, and intragenic variability [9]. To overcome these constraints, alternative phylogenetic gene markers like *tuf*, *secA*, and *secY* have been proposed [51,52]. In this context, genes encoding immunodominant membrane proteins, such as idpA (Immunodominant Protein A) and imp (Immunodominant Membrane Protein), offer promising alternatives for resolving phytoplasma diversity at a finer scale.

In the present study, we analyzed the genetic variability of the *imp* and *idpA* genes in phytoplasmas belonging to the 16SrIII group. Homologous sequences retrieved from the NCBI database facilitated the design of primers for the successful amplification of genomic fragments. Diversity analysis revealed that *imp* and *idpA* provide higher phylogenetic resolution than the 16S rRNA gene, with *imp* enabling a better classification at the subgroup level and *idpA* discriminating isolates within the same subgroup. This suggests that these genes could serve as complementary markers for phytoplasma classification. All novel field isolates analyzed in this study originate from Argentina, and although we have supplemented our data-set with publicly available genomes from the United States, Italy, Taiwan, and Chile, our sampling remains geographically constrained. The absence of isolates from other key regions in South America (e.g., Brazil, Paraguay, Uruguay, Peru) may bias estimates of subgroup prevalence and diversity and could influence the patterns of selection pressure we report. Future work should therefore prioritize pan-continental sampling especially across underrepresented countries to determine whether the genetic variability and evolutionary signals observed here for *imp* and *idpA* hold true throughout the broader 16SrIII lineage.

Selection pressure analyses indicated that the *imp* gene is under positive selection, with a general *dN*/*dS* value of 3.474 (*p* = 0.01) [44], suggesting adaptive evolution. Most positively selected sites were in the hydrophilic C-terminal region, which is exposed to the host environment and may be involved in host–pathogen interactions. This aligns with previous findings indicating that imp is highly variable among phytoplasmas, a characteristic often linked to its role in host adaptation and immune evasion [33]. Structural studies further support this hypothesis, showing that imp functions as an F-actin-binding protein, potentially influencing host cytoskeletal dynamics and facilitating infection [53]. The selective pressure on its extracellular domain reinforces the idea that imp plays a role in host recognition and transmission efficiency. Given its high variability, exposure to the host immune system, and functional relevance, *imp* emerges not only as a valuable phylogenetic marker [54] but also as a key determinant of phytoplasma pathogenicity.

In contrast, the *idpA* gene exhibited strong negative selection pressure (*dN*/*dS* = −3.090, *p* = 0.002) [44], with 115 out of 253 codons showing *dN*/*dS* > 0 values, predominantly in the hydrophilic extracellular region. This suggests that idpA is highly conserved, likely due to functional constraints essential for phytoplasma survival. Unlike imp, which undergoes strong positive selection and exhibits high variability, idpA appears to be under purifying selection, preserving its structure and function across different strains. Previous studies have shown that *idpA* expression levels vary across phytoplasma species. For example, in Western X-disease phytoplasma (Ca. Phytoplasma pruni, subgroup III-S), *IdpA* has been identified as the major immunodominant membrane protein [49], whereas in PoiBI Phytoplasma (Ca. Phytoplasma pruni, subgroup III-A), *imp* is more abundantly expressed, with *idpA* only detectable through immunohistochemistry but not by Western blot, likely due to lower expression levels [50]. These findings reinforce that *imp* and *idpA* are not homologous genes and that their relative expression differs among phytoplasma species, possibly reflecting distinct functional roles in host adaptation and transmission [33].

Phylogenetic analyses revealed that the 16S rRNA gene, though useful for broad classification, lacks resolution for distinguishing closely related phytoplasma strains [55]. In contrast, *imp* and *idpA* genes provided enhanced differentiation at both subgroup and intra-subgroup levels. For example, while 16SrIII-J isolates formed a monophyletic cluster based on 16S rRNA sequences, *imp* and *idpA* genes uncovered finer genetic differences, demonstrating their value for resolving intra-subgroup diversity. These findings are further supported by whole-genome analyses, which have shown that isolates from the 16SrIII-J subgroup, such as Cicuta witches’ broom phytoplasma (CicWB) and Vc33, form a distinct clade separate from other 16SrIII subgroups. This separation is strongly supported by genomic metrics such as Average Nucleotide Identity (ANI) and digital DNA–DNA hybridization (dDDH), which exceed 97% and 70%, respectively, within the 16SrIII-J subgroup [56]. These metrics, now integral to modern classification schemes for Candidatus Phytoplasma species [2,4], provide robust evidence for the distinct evolutionary trajectory of the 16SrIII-J subgroup. The correlation between gene-based phylogenies (*imp*, *idpA*) and whole-genome analyses underscores the importance of integrating multiple approaches to achieve a more accurate and comprehensive classification of phytoplasmas, particularly within highly diverse groups like the X-disease phytoplasmas (16SrIII). These findings highlight the potential of combining genomic and gene-specific markers to refine our understanding of phytoplasma diversity and evolution.

Notably, a similar pattern of hidden divergence within a ribosomal subgroup was observed in the MA-1 isolate, which, despite being classified as 16SrIII-B by 16S rRNA RFLP profiles, consistently formed a separate clade from other 16SrIII-B isolates (ChTDIII and CaesLL) based on *imp*, *idpA*, and whole-genome analyses [56]. This finding agrees with other reports across phytoplasma taxa, where strains indistinguishable by 16S RFLP require multilocus or genome-based approaches for accurate classification. For instance, [10] demonstrated that while 16SrV phytoplasmas (e.g., flavescence dorée strains) shared high 16S rRNA similarity, analysis of map and *uvrB*-*degV* genes was necessary to confirm their divergence into established subgroups like 16SrV-A (‘Ca. Phytoplasma ulmi’) and 16SrV-B (‘Ca. Phytoplasma ziziphi’). Similarly, [12] resolved taxonomic ambiguities in sugarcane phytoplasmas (16SrXI/XIV) by incorporating *leuS* and *secA* sequences, revealing that sugarcane whiteleaf and grassy shoot strains were identical, while Napier grass stunt formed a distinct lineage. In a parallel finding, [57] showed that 16SrIV-B and 16SrIV-D phytoplasmas were genetically identical across multiple loci (*secA*, *groEL*), despite prior 16S-based subgroup designations, and excluded the Tanzanian 16SrIV-C strain from the Caribbean clade. Together, these studies demonstrate that multilocus or genomic analyses are indispensable for precise strain differentiation. The divergence of MA-1 from other 16SrIII-B isolates therefore exemplifies the critical need to adopt more robust phylogenetic frameworks beyond single-gene analyses. Building on these phylogenetic insights, the *imp* gene demonstrates significant potential for developing diagnostic tools and disease management strategies. Its high intra- and interspecies variability makes it an excellent candidate for population and evolutionary studies, as demonstrated by [54], which identified 17 *imp* genotypes in ‘Ca. Phytoplasma pyri’, revealing its utility in distinguishing strains and understanding host adaptation. This variability also opens the door to the development of specific antisera or monoclonal antibodies targeting imp, which could be used for serological detection. For instance, [58] successfully developed an anti-Imp ELISA assay for detecting “flavescence dorée” phytoplasmas in grapevine, insect vectors, and host plants, demonstrating the feasibility of using imp-based serological tools for field diagnostics. Furthermore, the use of anti-Imp antibodies has been shown to improve phytoplasma genome sequencing efforts. The authors of [59] employed immunoprecipitation with anti-Imp antibodies to enrich phytoplasma DNA, enabling the assembly of high-quality genomes, such as that of ‘Ca. P. aurantifolia’ NCHU2014. In addition, [47] highlights the utility of the *imp* gene as a target for recombinase polymerase amplification (RPA) assays, which demonstrated comparable sensitivity to PCR for detecting ‘Ca. Phytoplasma pruni’ in sweet cherry tissues. Their findings revealed that *imp* is highly expressed in infected plants, with RNA transcript levels significantly higher than those of *idpA*, suggesting that *imp* may be the major immunodominant protein in this phytoplasma subgroup.

In summary, the study of *imp* and *idpA* genes has provided valuable insights into the diversity, evolution, and pathogenicity of phytoplasmas within the 16SrIII group. These findings not only enhance our understanding of phytoplasma biology but also pave the way for the development of more accurate diagnostic tools and effective management strategies.

## Figures and Tables

**Figure 1 microorganisms-13-01170-f001:**
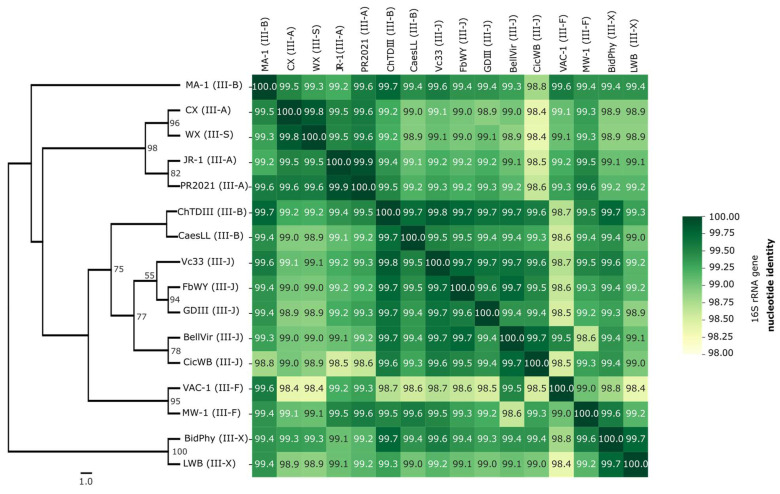
A phylogenetic tree and heatmap based on the 16S rRNA gene sequences of *X-disease* group phytoplasma strains. The dendrogram was inferred using the maximum likelihood method, and the bootstrap values (>50%) are indicated at the nodes. The accompanying heatmap shows pairwise nucleotide identity percentages among the strains. The color gradient reflects nucleotide identity ranging from 98.0% (light yellow) to 100.0% (dark green).

**Figure 2 microorganisms-13-01170-f002:**
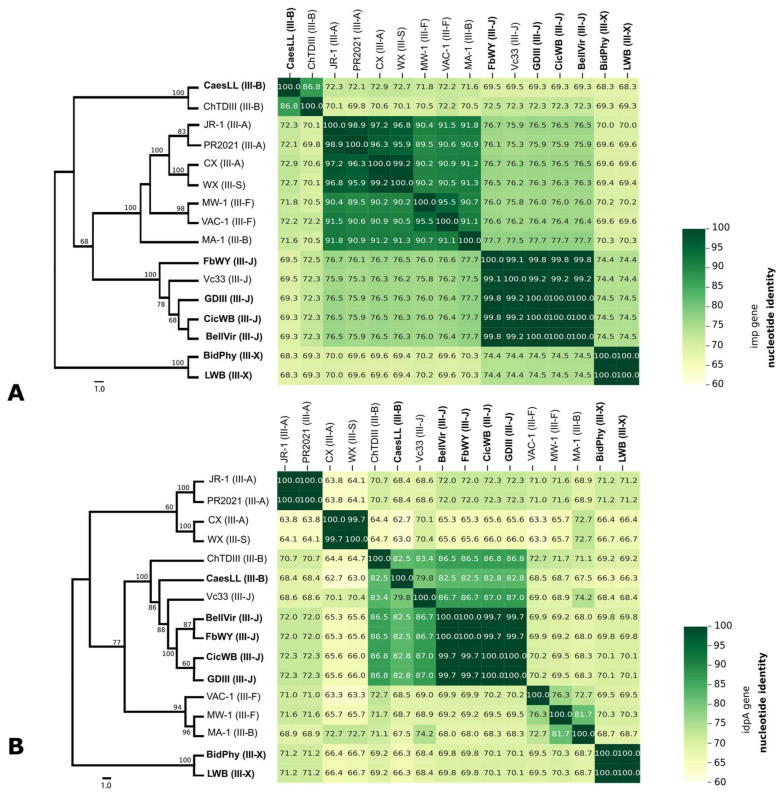
Phylogenetic trees and pairwise nucleotide identity matrices based on the imp (**A**) and idpA (**B**) gene sequences of X-disease group phytoplasmas. The heatmaps show the pairwise percentage of nucleotide identity, while the dendrograms represent the phylogenetic relationships inferred from the multiple sequence alignments of each gene using the ML method. The color gradients indicate identity values ranging from 63% (light yellow) to 100% (dark green). In bold are the sequences obtained in this work.

**Table 1 microorganisms-13-01170-t001:** List of strains used in PCR reactions to amplify *imp* and *idpA* ORFs. * 16SrIII subgroup based on 16S rRNA (1.2 kb) RFLP profiles, *imp* PCR/*idpA* PCR (number of samples tested/number of PCR positive samples); (-): NA.

Phytoplasma Strain	16SrIII *	Host	*imp* PCR (+)	*idpA* PCR (+)	#Accession (*imp*/*idpA*)
*Bellis* virescence (BellVir)	III-J	*Bellis perennis*	2/2	2/2	MG435348.1/MG435349.1
Garlic Decline (GDIII)	III-J	*Allium sativum*	2/2	2/2	PQ429243.1/PQ429237.1
Fodder Beet Wilting-Yellowing (FbWY)	III-J	*Beta vulgaris* var. *rapacea*	2/2	2/2	PQ429242.1/PQ429236.1
Sugar Beet Wilting-Yellowing (SugBeetWY)	III-J	*Beta vulgaris* var. altissima	0/3	0/3	-
Cicuta Witches Broom (CicWB)	III-J	*Conium maculatum*	2/2	2/2	PQ429241.1/PQ429238.1
China tree decline (ChTDIII)	III-B	*Melia azedarach*	3/3	3/3	NWN45603.1/NWN45596.1
Caesalpinia little leaf (CaesLL)	III-B	*Caesalpinia gilliesii*	2/2	2/2	PQ429239.1/PQ429233.1
Argentinean Peach Yellows (ArPY)	III-B	*Prunus persica*	0/3	0/3	-
Lettuce Witches’ Broom (LWB)	III-X	*Lactuca sativa*	2/2	2/2	PQ871563/PQ429235.1
Bidens Phyllody (BidPhy)	III-X	*Bidens subalternans*	1/3	1/3	PQ429240.1/PQ429234.1
Heterosperma Phyllody (HetPhy)	III-X	*Heterosperma ovatifolium*	0/3	0/3	-

**Table 2 microorganisms-13-01170-t002:** List of reference genomes used in this study. * 16S rRNA subgroup based on analysis in *i*Phyclassifier (https://plantpathology.ba.ars.usda.gov/cgi-bin/resource/iphyclassifier.cgi, accessed on 1 January 2025).

Phytoplasma [Strain]	16SrIII *	Host	Location	GenBank Accession
*Ca*. Phytoplasma pruni [WX]	III-S	*Prunus avium*	USA	AF533231.1
*Ca*. Phytoplasma pruni [CX]	III-A	*Prunus domestica*	USA	LHCF00000000.1
*Ca*. Phytoplasma pruni [PR2021]	III-A	*Euphorbia pulcherrima*	Taiwan	CP119306.1
Poinsettia branch-inducing [JR-1]	III-A	*Euphorbia pulcherrima*	USA	AKIK00000000.1
Clover Phyllody [MA-1]	III-B	*Chrysanthemum leuchantemum*	Italy	AKIM00000000.1
Vaccinium Witches’ Broom [VAC-1]	III-F	*Vaccinium myrtillus*	Italy	AKIN00000000.1
Milkweed Yellows [MW-1]	III-F	*Asclepias syriaca*	USA	AKIL00000000.1
*Ca*. Phytoplasma sp. [Vc33]	III-J	*Catharanthus roseus*	Chile	LLKK00000000.1
Chinaberry tree decline [ChTDIII]	III-B	*Melia azedarach*	Argentina	JABUOH000000000.1

**Table 3 microorganisms-13-01170-t003:** Selection pressure analysis in imp and idpA proteins, with 16S rRNA diversity metrics. N°: number of sequences; #nt: total nucleotide positions; S: segregating sites; P: nucleotide diversity (π); *dN*/*dS*: statistic test (NA for non-coding 16S); *p*-value: probability of rejecting strict neutrality; TM/HD: codons with *dN*/*dS* > 0 in transmembrane/hydrophilic domains; %: proportion of codons with *dN*/*dS* > 0; #codons: total codons (NA for 16S).

Dataset	N°	#nt	S	P	*dN*/*dS*	*p*-Value	Normalized *dN*/*dS* > 0			
							TM	HD	#Codons	%
imp	16	516	258	0.18272	3.474	0.01	10	70	157	50.955
idpA	16	857	268	0.10407	−3.090	0.002	11	104	253	45.454
16S rRNA	16	1163	94	0.01354	NA	NA	NA	NA	NA	NA

## Data Availability

Nucleotide and Aminoacidic sequence from imp and idpA membrane proteins obtained in this paper were deposited in the NCBI repository under the accession numbers PQ429233 to PQ429243 and PQ871563.

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
