# Peer review of "Genetic Diversity in Phytoplasmas from X-Disease Group Based in Analysis of idpA and imp Genes"

_microorganisms, 2025, doi:10.3390/microorganisms13051170_

Round 1
Reviewer 1 Report
Comments and Suggestions for Authors
This manuscript presents the genetic diversity of phytoplasmas associated with X-disease, based on the idpA and imp genes. Given the urgent challenges in developing reliable methods for classifying phytoplasmas while enabling finer strain characterization from an evolutionary perspective, the authors offer a solid approach to begin addressing these issues using strains from group 16SrIII, which are widespread in South America. The manuscript is well-written. I have provided specific comments on key areas for improvement in the attached PDF.

Author Response
Response to reviewer 1
We thank the reviewer for their thoughtful and constructive feedback, which helped us improve the clarity and contextual depth of the manuscript. Below we provide a point-by-point response to each comment.
Comment #1: I suggest to clarify this sentence more precisely. The threshold is a new proposal and another one was "traditionally" used (lanes 31-32)
Response#1: The reviewer correctly notes that the 98.65% threshold is a 2022 revision (not "traditional"), and requests clarification of the historical vs. current classification systems. In the revised MS a revision of the flagged sentence, with added context to distinguish between older RFLP-based grouping and modern sequence-based thresholds are added.
Comment #2: please list some of those and link them with the purpose of the present study
Response #2: We appreciate your comment. As suggested, we revised the manuscript text to clearly state the limitations of the 16S-based classification system and our alternative marker approach for improved clarity (lanes 39-54)
Comment #3: There are various papers that support this point for other group of bacteria. The other should cite some of them here.
Response #3: We thank the reviewer for the helpful suggestion. In response, we have cited additional studies that support the shift toward genome-based taxonomy in other bacterial groups. Specifically, we now reference García-López et al. (2019), who demonstrated improved taxonomic resolution through the analysis of over 1,000 type-strain genomes of Bacteroidetes, and Riesco & Trujillo (2024), who provide updated standards for the use of genome data in prokaryotic taxonomy. Accordingly, the reference list has been adjusted to maintain continuous numerical order, as required by the journal's formatting guidelines.
Commnet #4: why 16SrIII-X has not been included? Please provide a justification here.
Response#4: We thank the reviewer for this observation. We believe there may have been a misunderstanding regarding the inclusion of 16SrIII-X, which was indeed analyzed in this study, as stated in the aims and detailed in the Materials and Methods and Table 1. It is possible that the reviewer was referring to the 16SrIII-W subgroup instead. We now clarify in the Materials and Methods section that although 16SrIII-W was initially considered, we were unable to amplify any DNA from those samples, not even the 16S rRNA gene, despite multiple attempts with different extraction and PCR protocols. For this reason, 16SrIII-W was excluded from the analysis.
Comment #5: Is this belonging here or below? (referring to the title of section 2.3: Cloning and PCR amplification of imp and idpA genes)
Response #5: After reconsidering the structure, we agree that including “cloning” in the title of section 2.3 might be misleading, as the actual cloning steps are described in the following section (2.4). To improve clarity and better reflect the content of each section, we have modified the title of section 2.3 to “PCR amplification of imp and idpA genes” and kept the cloning and sequencing steps exclusively under the revised title of section 2.4: “Cloning and sequencing of imp and idpA genes”. We believe this change enhances the logical flow of the methodology.
Comment # 6: please add Country and host in parentheses (lanes 181 and 185)
Response #6: Done
Comment #7: These colors may be very confusing for color-blind people. Please try to find a more inclusive strategy to indicate the boxes (referred to Figures 1-2).
Response #7: We have taken your suggestion into account and made the necessary adjustments to the figures by replacing the color-based indicators with lines and dotted lines. We believe this modification improves clarity and inclusivity, and we hope it will now be easier for all users to interpret.
Comment #8: referred to dN-dS > 0 in idpA analysis
Response #8: Thank you for your comment. In our analysis, we focused on dN-dS values greater than 0 because we were specifically looking for evidence of positive selection pressure. However, in the case of the IdpA protein, although 50 codons showed negative dN-dS values, the overall analysis suggests that the protein is under negative selection pressure. We believe that modifying the text slightly to reflect this distinction may improve the clarity of the results (see lanes 225-233)
Reviewer 2 Report
Comments and Suggestions for Authors
The manuscript titled "Genetic Diversity in Phytoplasmas From X-Disease Group Based in Analysis of idpA and imp Genes" presents a significant contribution to the understanding of phytoplasma diversity, particularly through the analysis of idpA and imp genes. This study is well-structured and addresses important gaps in the current literature. However, a few comments can enhance clarity and impact.
- A comparative analysis of other known phytoplasma genes could strengthen this manuscript. This would help contextualize the findings within the broader landscape of phytoplasma research and highlight the significance of the idpA and imp genes.
- The addition of phylogenetic trees/graphs illustrating selection pressures would enhance the clarity of the results.
- This manuscript would benefit from a more thorough discussion of its limitations, particularly regarding the geographic scope of the samples and how this may influence the findings.
- This would enhance the manuscript's contribution to the field of the differences between idpA and imp, and a more comprehensive comparative analysis with other known phytoplasma genes could provide additional context and support for the findings.
Author Response
Response to Reviewer 2
We thank the reviewer for their thoughtful and constructive feedback, which helped us improve the clarity and contextual depth of the manuscript. Below we provide a point-by-point response to each comment.
Comment#1: "A comparative analysis of other known phytoplasma genes could strengthen this manuscript. This would help contextualize the findings within the broader landscape of phytoplasma research and highlight the significance of the idpA and imp genes."
Response#1:
We fully agree that broader comparative analyses using additional genes (e.g., secY, rp, tuf, leuS) can enrich phytoplasma diversity studies. However, incorporating these markers for our specific set of field isolates would require additional PCR primer development and sequencing, which goes beyond the experimental scope and resources of the present study.That said, we have added new text in the Introduction (lines 42-54, 86-96) referencing relevant studies where these markers have been used for phytoplasma classification. Furthermore, we note that while such multilocus approaches are increasingly applied in well-characterized groups like 16SrI (aster yellows), the 16SrIII group currently lacks the same breadth of genomic data and validated markers. This underscores the importance of exploring alternative single-gene markers like imp and idpA, which are broadly distributed, functionally relevant, and useful for differentiating 16SrIII subgroups.
Comment#2: "The addition of phylogenetic trees/graphs illustrating selection pressures would enhance the clarity of the results."
Response#2: We appreciate this suggestion and agree that visualizing the results of selection pressure analysis can enhance interpretability. As requested, we have now included a new supplementary figure (Figure S2) showing codon-by-codon dN-dS values across the imp and idpA genes.
Comment#3: "This manuscript would benefit from a more thorough discussion of its limitations, particularly regarding the geographic scope of the samples and how this may influence the findings."
Response#3: Thank you for pointing this out. In the revised Discussion section (lines 438–445), we have added a paragraph explicitly addressing the geographic scope of our sampling.
Comment: "...a more comprehensive comparative analysis with other known phytoplasma genes could provide additional context and support for the findings."
Response:
As noted in our response to the first comment, we agree that integrating additional markers can provide broader phylogenetic context. While experimental limitations prevent us from including sequence data for other protein-coding genes in this study, we have expanded the text to discuss how imp and idpA complement existing markers. We also emphasize that within the 16SrIII group, available genomic data and marker validation remain relatively limited compared to better-characterized groups like 16SrI. This highlights the relevance of our findings and the utility of membrane protein genes as robust classification tools within this underexplored group.
Reviewer 3 Report
Comments and Suggestions for Authors
This interesting paper is devoted to the use of two novel molecular markers - imp and idpA genes - to reveal phylogenetic diversity and evolutionary history of X-disease phytoplasmas. Indeed, multilocus phylogeny has been among the most powerful tools for investigating taxonomy and diagnostics of plant pathogens. So, the paper looks timely and scientifically important. At the same time, there are some points need to be improved since it can be accepted for publication.
Major comments
- The authors used 8 Phytoplasma strains and 11 sequences retrieved from databases to perform phylogenetic study. I believe this number is too low to make conclusions about phylogenetic potential of markers used. At the same time I understand that information on Phytoplasma X-disease genomes is extremely restricted. Can the authors add a bit more strains in the study? (This note is optional)
- I think more phylogenetic information is needed. In the Introduction, the authors said that secY, rp and leuS genes have also been used for Phytoplasma's phylogenetic studies. Why don't the authors apply these markers in this study? I think it would be interesting to compare more barcodes and individual phylogenies to elucidate relationships between strains tested.
- In addition to individual phylogenetic trees, represented in Figs. 3-4, a concatenated tree constraucted based on 16S rDNA + idpA + imp would be important. Also, the Table 3 should contain data on phylogenetic properties of 16S rDNA sequences also.
- I think the authors should discuss in more detail the MA-1 sequence and the fact that, despite being assigned to 16Sr-III B group, it forms a clearly distinct clade in 16S rDNA and a sepatare clade in imp tree. How these differences can be explained? Minor comments:
- Line 75: 'Recombinant'
- Line 92: 'strains'
- Chapter 2.6. The authors should note what algorithm was used to construct phylogenetic trees (as I can see, it was ML method)
- Chapter 3.2. Probably, the primes sequences should be moved to 'Cloning and PCR amplification of imp and idpA genes'
- Lines 169-171: fragment sizes were 522-546 for both markers?
- Figures 1,2, Probably, these figures could be moved to Supplementary
- Line 237: I do not see 100 % bootstrap support for this group, I see only 55-94 % bootstrap support values.
- Figure 3: 'LWB', not 'LWD'
- Line 274: 'Putative'
Author Response
Responses to Reviewer 3
We are grateful to the reviewer for the encouraging assessment and valuable suggestions. We address each point below and have made corresponding modifications to the manuscript to enhance its scientific rigor and clarity.
Major Comments
Comment 1:
“The authors used 8 phytoplasma strains and 11 sequences retrieved from databases... I believe this number is too low to make conclusions about phylogenetic potential of markers used. Can the authors add a bit more strains in the study? (This note is optional).”
Response:
We appreciate this observation and agree that a larger dataset could further strengthen phylogenetic inferences. However, as noted by the reviewer, publicly available genome sequences of X-disease group phytoplasmas remain scarce, particularly for well-characterized representatives of subgroups beyond 16SrIII-A. Within this constraint, we have maximized our dataset by including all complete and draft genomes available in public repositories (see Table 2), as well as sequences from eight well-characterized Argentine isolates.We have now clarified this limitation and its implications in the revised Discussion , and also highlight the need for continued genomic efforts to expand the reference database for this important group.
Comment 2:
“Why don’t the authors apply secY, rp, and leuS markers in this study? I think it would be interesting to compare more barcodes and individual phylogenies.”
Response:
Thank you for this insightful comment. While we agree that including additional markers such as secY, rp, and leuS could be informative, experimental analysis of these genes would require further PCR optimization and sequencing for all our field isolates, which is beyond the current study's scope.Nevertheless, we have now expanded the Introduction and Discussion to reference prior studies using these and other markers, and we compare their utility to that of imp and idpA.
Comment 3:
“In addition to individual phylogenetic trees, a concatenated tree (16S rDNA + idpA + imp) would be important. Also, Table 3 should contain data on phylogenetic properties of 16S rDNA.”
Response:
We thank the reviewer for these valuable suggestions. In response:
We have added a concatenated phylogenetic tree (16S rDNA + idpA + imp) as Supplementary Figure S3, reconstructed using maximum-likelihood criteria. This combined analysis strengthens our subgroup delineations while maintaining consistency with the individual gene trees (Figures 3-4).For Table 3, we have incorporated 16S rDNA phylogenetic properties (total aligned positions, segregating sites, and nucleotide diversity π) to enable direct comparison with protein-coding genes. While selection parameters (dN-dS) are not applicable for this non-coding marker, these nucleotide-level metrics provide important insights into sequence conservation patterns.The Results section (3.3) now discusses these integrated findings, and we have updated the Materials and Methods to reflect both the concatenated analysis and 16S rRNA diversity calculations. These modifications provide a more comprehensive view of both evolutionary relationships and sequence variability across all loci.
Comment 4:
“The authors should discuss in more detail the MA-1 sequence, which, despite being 16SrIII-B, forms a separate clade in 16S rDNA and imp trees...”
Response:
We fully agree with the reviewer's observation regarding the phylogenetic divergence of MA-1. We have addressed this point in the revised Discussion (lines 413-431), where we now provide a detailed analysis of MA-1's unique position in both the 16S rDNA and imp phylogenies, despite its classification as 16SrIII-B, and discuss the potential implications of this discrepancy for phytoplasma classification
Minor Comments
-Line 75 'Recombinant':
-Response: Corrected.
-Line 92 – 'Strains':
-Response: Corrected.
-Chapter 2.6 – Clarification on the algorithm used:
-Response: Added a sentence clarifying that all phylogenetic trees were constructed using the maximum-likelihood (ML) method implemented in IQ-TREE.
Chapter 3.2 – Primer sequences relocation:
Response: As suggested, we have moved the primer sequences to section 2.3 (“Cloning and PCR amplification of imp and idpA genes”) to improve logical flow.
Lines 169–171 – Fragment sizes were 522–546 for both markers?:
Response: We thank the reviewer for catching this point. Upon further clarification, we confirm that the fragment size ranges differ between the two genes. For the imp gene, the open reading frame lengths ranged from 522 to 546 bp among the strains studied. In contrast, the idpA gene showed greater size variation, ranging from 864 bp (e.g., CX isolate) to 987 bp (e.g., JR1, WX, and MW1 isolates). We have revised this section in the manuscript accordingly to accurately reflect these differences .
Figures 1 & 2 – Possibly move to Supplementary:
Response: We appreciate the reviewer’s suggestion and understand the rationale for streamlining the main text. However, we respectfully prefer to retain Figures 1 and 2 in the main manuscript. These figures illustrate the conserved and variable domains of the imp and idpA proteins across representative phytoplasma strains and directly support the key findings regarding domain-specific selection pressures discussed in the Results and Discussion sections.We believe that having these figures readily accessible in the main text enhances the reader's ability to follow the molecular basis for the observed selection patterns and phylogenetic divergence. As such, we feel that their inclusion contributes substantively to the clarity and impact of the manuscript.
Line 237 – Bootstrap value for 16SrIII-J group not 100%:
Response: Correct observation. The sentence has been corrected to accurately report the bootstrap range observed.
Figure 3 – 'LWB', not 'LWD':
Response: Typo corrected. Also the Figure 3 was improved and new version was incorporated to revised MS.
Line 274 – 'Putative':
Response: Typo corrected.
Round 2
Reviewer 3 Report
Comments and Suggestions for Authors
In the revised version of the paper the authors improved almost all the points mentioned in the review. Now, the manuscript looks more complete and comprehensively describing the research topic. I believe the revised version can be accepted, but a couple of points should be taken into account:
- I think the Lines 53-54 and 104-106 duplicate each other. Probably, one of these sentences should be removed.
- Among the immunodominant membrane proteins, the authors mentioned Amp gene also. Why this gene was not used for the analysis?
- I think the labels in Fig. 2 are very hard to be written and understtod. Can the authors divide the illustration into two parts and put them under each other? Or, probably, this figure can be moved to Supplementary and present in album orientation? (optional)
Author Response
Response to reviewers MS round 2
Comments and Suggestions for Authors:
In the revised version of the paper the authors improved almost all the points mentioned in the review. Now, the manuscript looks more complete and comprehensively describing the research topic. I believe the revised version can be accepted, but a couple of points should be taken into account:
Comments_1: I think the Lines 53-54 and 104-106 duplicate each other. Probably, one of these sentences should be removed.
Response 1: We thank the reviewer for pointing out the redundancy in the Introduction. We have revised and streamlined the final section to avoid repetition, combining the description of immunodominant membrane proteins and the study's aims into a single, coherent paragraph.
Comments_2: Among the immunodominant membrane proteins, the authors mentioned Amp gene also. Why was this gene not used for the analysis?
Response 2: We appreciate the reviewer’s comment. In response, we clarify that we also searched for the presence of the amp gene in the analyzed genomes, using the amp protein sequence of 'Candidatus Phytoplasma asteris' strain AYWB (GenBank accession number:WP_011412878.1 ) as a query. However, no amp homologs were identified in any of the X-disease group genomes analyzed. We have added this information to the Results section for clarification.
Comments_3: I think the labels in Fig. 2 are very hard to be written and understtod. Can the authors divide the illustration into two parts and put them under each other? Or, probably, this figure can be moved to Supplementary and present in album orientation? (optional)
Response_3: Thank you for your comment. In response, we will move Figures 1 and 2 to the Supplementary section, with Figure 2 presented in landscape orientation. Additionally, the figures in the text have been renumbered to reflect these changes.